# Acidic Shift of Optimum pH of Bovine Serum Amine Oxidase upon Immobilization onto Nanostructured Ferric Tannates

**DOI:** 10.3390/ijms232012172

**Published:** 2022-10-12

**Authors:** Graziano Rilievo, Alessandro Cecconello, Simone Molinari, Andrea Venerando, Lavinia Rutigliano, Gayathri T. Govardhan, Dinusha H. Kariyawasam, Ruth J. Arusei, Lucio Zennaro, Maria L. Di Paolo, Enzo Agostinelli, Fabio Vianello, Massimiliano Magro

**Affiliations:** 1Department of Comparative Biomedicine and Food Science, University of Padua, Viale dell’Università 16, 35020 Legnaro, Italy; 2Department of Geosciences, University of Padua, Via Gradenigo 6, 35131 Padova, Italy; 3Department of Molecular Medicine, Sapienza University of Rome, Viale Regina Elena 291, 00161 Rome, Italy; 4Department of Sensory Organs, Sapienza University of Rome, Policlinico Umberto I, Viale del Policlinico 155, 00161 Rome, Italy; 5Department of Molecular Medicine, University of Padua, via Gabelli 63, 35121 Padova, Italy; 6International Polyamines Foundation ‘ETS-ONLUS’, Via del Forte Tiburtino 98, 00159 Rome, Italy

**Keywords:** enzyme–nanoparticle hybrids, bovine serum amine oxidase, enzyme activity, pH dependence, protein nano-immobilization, NMR relaxometry, polyamines

## Abstract

Protein–nanoparticle hybrids represent entities characterized by emerging biological properties that can significantly differ from those of the parent components. Herein, bovine serum amine oxidase (i.e., BSAO) was immobilized onto a magnetic nanomaterial constituted of surface active maghemite nanoparticles (i.e., SAMNs, the core), surface-modified with tannic acid (i.e., TA, the shell), to produce a biologically active ternary hybrid (i.e., SAMN@TA@BSAO). In comparison with the native enzyme, the secondary structure of the immobilized BSAO responded to pH variations sensitively, resulting in a shift of its optimum activity from pH 7.2 to 5.0. Conversely, the native enzyme structure was not influenced by pH and its activity was affected at pH 5.0, i.e., in correspondence with the best performances of SAMN@TA@BSAO. Thus, an extensive NMR study was dedicated to the structure–function relationship of native BSAO, confirming that its low activity below pH 6.0 was ascribable to minimal structural modifications not detected by circular dichroism. The generation of cytotoxic products, such as aldehydes and H_2_O_2_, by the catalytic activity of SAMN@TA@BSAO on polyamine oxidation is envisaged as smart nanotherapy for tumor cells. The present study supports protein–nanoparticle conjugation as a key for the modulation of biological functions.

## 1. Introduction

Enzymes are increasingly used as effective biopharmaceuticals with potential applications in a wide range of diseases [1]. Tumor proliferation is frequently correlated with polyamine overproduction and the expression and activity of enzymes involved in the biosynthesis of these substances [2]. For alternative strategies for the inhibition of proliferation of cancerous cells, amine oxidases represent an attractive option [3], catalyzing in situ production of cytotoxic molecules (i.e., aldehydes and hydrogen peroxide) through the oxidative deamination of polyamines:(1)$$R-CH_{2}NH_{2}+O_{2}+H_{2}O\;\overset{Amine\;oxidase}{\to }\;R-\mathrm{CHO}\;+\;NH_{3}+H_{2}O_{2}$$

Polyamine concentration in the target malignant tumor mass, which is often significantly higher than in normal cells [4], would induce apoptosis in the presence of amine oxidases.

Copper-containing amine oxidases are key regulators of polyamine metabolism, and they are present in relatively high concentrations in ruminant serum [5,6]. Among them, bovine serum amine oxidase (BSAO, EC 1.4.3.6) was already successfully isolated [7,8] and tested for inducing cytotoxicity in human tumor cells [3,9,10], overcoming the issue of their multidrug-resistant (MDR) counterpart [11], and thus it is a potential candidate for future therapeutic applications.

Unfortunately, the applicability of BSAO in real-world scenarios suffers from limitations related to the use of enzymes as drugs, including their very low membrane permeability and intrinsic instability [12]. In addition, the catalytic activity of enzymes is strongly pH-dependent, and can be lost even at pH values not so far from their optimum [13,14]. In the case of BSAO, this reduction in activity was attributed to protonation/deprotonation phenomena of amino acid side chains involved in the catalysis [15].

Nanomaterial-based delivery strategies can cope with all these limitations, providing smart nanovehicles that can be ideally targeted and accumulated into neoplastic tissue [16]. Recent studies have demonstrated that the immobilization of an enzyme onto a nanoparticle can improve macromolecule stability [17] and even enhance the activity, specificity, and selectivity of the immobilized biocatalyst when compared to the soluble enzyme [18]. Nevertheless, successful enzyme–nanomaterial hybridization remains far from being a trivial task, requiring suitable nanomaterial surfaces for harboring the enzyme, as well as smart binding strategies to avoid the well-known immobilization-related risks, such as protein denaturation [19,20,21,22].

Among nanomaterials, bare iron oxide nanoparticles exhibit appealing properties, including magnetic, physical, and biochemical characteristics, that are useful for their application in several fields [23].

Aiming at a general approach for protein immobilization onto bare iron oxide nanoparticles, we used tannic acid (i.e., TA) as an ideal precoating material for promoting macromolecule docking [24,25]. This choice was due to TA’s ability to interact with the crystalline structure of different iron oxides [26] and its high affinity for proteins [27]. TA has attracted widespread interest as a promising natural compound to design advanced materials, with applications including tissue engineering, treatment of infections, cancer therapy, biosensing, and drug delivery [28]. In drug delivery, for example, TA was employed as a molecular “glue” for mediating the interaction of nanomaterials with drug molecules and polymers [26].

In this context, the general solution of surface-active maghemite nanoparticles (i.e., SAMNs) modified with tannic acid (i.e., SAMN@TA) [25] is proposed for the generation of a biologically active BSAO nanobioconjugate.

## 2. Results

Aiming at the production of a biologically active ternary hybrid, BSAO was directly bound onto the SAMN@TA (i.e., SAMN@TA@BSAO). The details regarding the assembly strategies, the reagents used, and additional experimental data are described in the Materials and Methods section and in the Appendix A. In Figure 1, the SAMN@TA@BSAO-catalyzed oxidation of polyamines is schematically represented. This enzymatic reaction can be ideally exploited for the in situ production of cytotoxic substances (aldehydes and hydrogen peroxide) in acidic and polyamine-rich environments, such as tumor tissue [29].

### 2.1. Physical–Chemical Characterization of the SAMN@TA@BSAO Hybrid

The direct immobilization of BSAO onto SAMN@TA was studied by plotting adsorption isotherms as per Giles et al. [30] and Langmuir [31] models. Measurements were carried out at constant SAMN@TA concentration (500 mg L^−1^) as a function of BSAO concentration in the 25–200 mg L^−1^ range. The Giles model [30] is a useful preliminary approach that considers the shape of the curve of the adsorbate (*Q*) against the free ligand at the equilibrium (*C_e_*). Herein, the system displayed a saturation behavior that indicated that once the first BSAO shell is completed, no further BSAO adsorption to SAMN@TA can take place. This results in the formation of a molecular monolayer (Appendix A). On these bases, the Langmuir isotherm model represents the proper model for a detailed study of the current system. Actually, Langmuir assumes the formation of a single adsorbate shell and the availability of equivalent, independent binding sites for the ligand on the material surface [31]. The Langmuir isotherm equation was used in the following linearized form:(2)$$\frac{1}{Q}=\frac{1}{Q_{max}}+\frac{1}{Q_{max}\;K_{L}\;C_{e}}$$
where *Q* is the surface concentration of BSAO bound to SAMNs, expressed in terms of mass ratio (mg BSAO g^−1^ SAMN@TA), *Q_max_* is the maximum adsorption capacity, *K_L_* is the Langmuir adsorption constant, and *C_e_* is the equilibrium concentration of BSAO in solution (mg L^−1^). The values of *K_L_* and *Q_max_* for the binding of BSAO onto SAMN@TA were calculated to be 6.5 mL mg^−1^ and 91 mg g^−1^, in good agreement with the protein loading reported in previous studies [32] (Appendix A). The morphological and hydrodynamic properties of the hybrid nanomaterials were characterized by transmission electron microscopy (TEM) and dynamic light scattering (see Materials and Methods section). TEM micrographs of SAMN@TA (Figure 2a) showed monodispersed core–shell spherical nanostructures. The presence of an organic phase, characterized by a lower electron density and a thickness of about 1.3 nm (blue line, Figure 2b), is clearly recognizable and ascribable to the TA layer. Besides the carbonaceous shell, the integrity of the SAMN magnetic core is evidenced. Furthermore, the spontaneous immobilization of BSAO can be detected by a significant increase in the organic phase following the self-assembly reaction of the enzyme with SAMN@TA. However, the relatively contained magnitude of the overall organic shell thickness, measuring 2.3 nm, can be explained considering the denaturation conditions induced by the TEM sample preparation (Figure 2c). In addition, SAMN@TA and SAMN@TA@BSAO were compared to bare SAMNs by X-ray diffraction (XRD) and scanning electron microscopy–energy-dispersive X-ray (SEM-EDX), further substantiating the conservation of the maghemite nanoparticles and confirming the formation of the organic envelopes (Appendix A).

The zeta potential (ζ) values of naked SAMNs, SAMN@TA, and SAMN@TA@BSAO hybrids were characterized. In comparison to bare SAMNs, the zeta potential values were strongly affected by the outermost layers of the hybrids, further confirming the occurrence of the sequential self-assembly reactions. The zeta potential of naked SAMNs was +39 ± 1 mV (conductivity = 0.004 mS/cm in water at 22 °C), which was attributed to positively charged coordination water molecules stabilizing undercoordinated Fe (III) sites on the nanoparticle surface [32]. After TA-SAMNs self-assembly reaction, the zeta potential drastically changed from being highly positive to −32 ± 1 mV (conductivity = 0.019 mS/cm in water at 22 °C). This is due to the complexation by tannic acid of the aforementioned iron sites, shielding the former bare surface of SAMNs. In addition, the anionic nature of tannic acid contributes additional negative charges, producing an overall negative electric potential at the solvent interface. The zeta potential of SAMN@TA was further altered by the binding of BSAO. Upon enzyme immobilization on the surface of SAMN@TA, the zeta potential shifted to −21 ± 1 mV (conductivity = 0.004 mS/cm in water at 22 °C). The change in the negative potential with respect to SAMN@TA may be attributed to the integration of the zwitterionic protein onto the complex.

The analysis of the hydrodynamic “radii” is reported in Figure 2d. For naked SAMNs, the hydrodynamic size was 184 ± 5 nm (Figure 2d, orange bars), which, upon hybridization with tannic acid, registered an increase to 307 ± 1 nm (Figure 2d, blue bars). Finally, upon BSAO binding, a further size enhancement was reported, resulting in a final hybrid size of 753 ± 8 nm (Figure 2d, green bars). The magnitude of the measured increase of the hydrodynamic radius is compatible with the conjugation of a large macromolecule, such as BSAO, displaying a well-preserved tertiary structure.

It is largely accepted that physical–chemical properties of colloidal nanoparticles are strongly influenced by their local environment. This differs significantly from the bulk, due to the interactions occurring at the nanoparticle–solvent interface [33]. Hence, in order to gather insights on the effect of bulk pH on the hybrid nanoenvironment, the ζ-potential value of SAMN@TA@BSAO ternary hybrid was determined in the 4.0–8.0 pH interval (Figure 3).

As a control, the SAMN@TA core–shell nanostructure was examined, revealing an almost unperturbed large negative ζ potential in the whole range (Figure 3, black squares). This noticeable feature was already reported for other TA-modified nanomaterials [34]. Differently, the behavior of the SAMN@TA@BSAO ternary hybrid showed less negative zeta-potential values, which tended to become more negative at lower pH values, indicating that the electrostatics of the hybrid are influenced by the outermost protein layer (Figure 3, red dots). Noteworthy, the point of zero charge of the ternary hybrid was at pH = 3.5, well below the isoelectric point of BSAO (pH = 6.1–6.2) [8]. This is ascribable to the TA layer conferring a polyanionic character to the ternary hybrid, as already observed for the conjugation of trypsin with tannic acid-modified Fe_3_O_4_ nanoparticles [34].

Fourier-transform infrared (FTIR) spectroscopy was employed to confirm the successful BSAO binding to SAMN@TA and to evidence possible structural modifications of the enzyme following its integration into the ternary hybrid [35]. The infrared spectrum of the SAMN@TA hybrid was dominated by features arising from the ferric tannate layer (Figure 4A). In particular, besides the three main maghemite vibrations at 555, 633 and 692 cm^−1^, a series of bands in the 1000–1710 cm^−1^ region were visible (Figure 4A, black and blue curves) representing C-O and C=O symmetric stretching and C=C vibrations of the aromatic rings [36,37,38].

The FTIR spectrum of native BSAO (Figure 4A, green curve) shows the typical amide I feature at 1633 cm^−1^, representing the C=O stretching vibration of the peptide bond, and the amide II band at 1547 cm^−1^, due to in-plane NH bending and CN stretching of the peptide bonds. The spectral profile of SAMN@TA@BSAO presents three main vibrations in the 500–700 cm^−1^ region related to the Fe-O stretching arising from the metal oxide lattice and a broad band centered at 3330 cm^−1^ deriving from the hydroxyl stretching vibrations [39]. Moreover, the spectral features of SAMN@TA are well preserved, as in the case of the double band in the 1300–1400 cm^−1^ region, characterized by two maxima at 1318 and 1435 cm^−1^ and the band at 1069 cm^−1^, which can be assigned to C-O stretching vibrations of tannic acid [36,37,40]. Besides the expected maghemite and tannic acid vibrations, amide I and II bands are clearly recognizable at 1654 cm^−1^ and 1574 cm^−1^. In this view, it is important to recall that amide I and amide II bands represent the FTIR fingerprint of native proteins [41], showing successful enzyme immobilization on the modified nanomaterial and suggesting the preservation of the enzyme tertiary structure. The amide I bands of native BSAO and of SAMN@TA@BSAO ternary hybrid were subjected to deconvolution according to Hebia and coauthors [42] for the determination of secondary structure composition (Figure 4B,C). The content of each structural component is reported in Table 1. In agreement with literature [43], besides a high content of undetermined structures, the deconvolution process showed that the secondary structure of BSAO is mainly characterized by β-strands, with a limited contribution of α-helices (≈10%).

Interestingly, the binding of BSAO to SAMN@TA was accompanied by a decrease (−50%) of α-helix content counterbalanced by an increase of β-sheet (+20%). The attenuation of the α-helical and the increase in β-sheet content can be seen as a conformational modification due to the interaction of the macromolecule with the nanoparticle surface, possibly resulting in the modification of the protein functional characteristics [44]. Moreover, the modification of protein conformation from α-helix to β-sheet was already interpreted as an enhancement of protein compactness [45,46,47]. No significant differences in other secondary structures emerged from the comparison of the native and nanoparticle-bound enzyme.

### 2.2. Comparison of the Activity and Structure of Native BSAO and of SAMN@TA@BSAO Hybrid as a Function of pH

The kinetic parameters (i.e., *k_cat_* and *K_M_*) of native and SAMN@TA-immobilized BSAO were determined and compared. The rate of the catalyzed oxidation of spermine by the two enzyme forms was measured as a function of pH in the 4.0–8.0 range. The pH dependence of *k_cat_* values were plotted according to Koudelka-modified Tipton and Dixon model described in Appendix A [48,49,50], and is shown in Figure 5.

The comparison of the two curves evidences in both cases a bell-shaped behavior indicating the involvement of two ionizable groups in the catalysis, as already described by Di Paolo et al. [15]. In particular, the two ionizable groups showed a pK_a,1_ at 7.3 ± 0.5 and pK_a,2_ at 7.8 ± 0.5 for the native enzyme, while for the immobilized form pK_a,1_ was 4.6 ± 0.5 and pK_a,2_ = 5.2 ± 0.5.

The *k_cat_* values associated with the SAMN@TA@BSAO hybrid, even if lower than that of the native enzyme (*k_cat_* 1.2 min^−1^ and 20 min^−1^, respectively), indicated that SAMN@TA generates a favorable local environment for enzyme immobilization, avoiding any substantial loss of the three-dimensional structure of BSAO, as supported by the FTIR and CD analysis (see hereafter). A shift in the optimum pH was observed upon enzyme immobilization. While native BSAO showed the highest catalytic constant at pH = 7.2, the SAMN@TA@BSAO hybrid displayed the highest value at pH = 5.0, steadily decreasing at higher pH values.

Comparison of pH dependences of *K_M_* values of native and SAMN@TA-bound enzyme forms on spermine showed differences as well. Differently from native BSAO, which presents a strong effect of pH on the Michaelis constant [15], *K_M_* values of the immobilized enzyme were limited to 0.2–6.0 µM in the whole pH range explored (Appendix A). We analyzed the catalytic efficiency (*k_cat_*/*K_M_*), which represents the second-order rate constant of the enzyme-catalyzed reaction and describes the kinetics at low substrate concentration, namely, when the enzyme is not saturated by substrate. In agreement with previous studies [15], the measured *k_cat_*/*K_M_* value of the native enzyme for spermine was strongly dependent on pH, decreasing by several orders of magnitude as the pH changed from neutral to acidic conditions (Δlog(*k_cat_*/*K_M_*)/ΔpH > 2 in the 5.0–7.2 pH range). Conversely, the catalytic efficiency of the SAMN@TA-immobilized enzyme towards spermine resulted in only slightly dependence on pH, increasing under acidic conditions (Δlog(*k_cat_*/*K_M_*)/ΔpH ≈ −0.27 in the 4.0–6.0 pH range) and almost insensitive at pH above 6.0 (Δlog(*k_cat_*/*K_M_*)/ΔpH ≈ 0.1 in the 6.0–8.0 pH range) (Appendix A).

The different pH sensitivity and the dramatic shift in optimum pH upon enzyme immobilization could be interpreted by recalling the concept of “nanoenvironment.” That is, the local pH experienced by the immobilized enzyme is less acidic than that of native protein in solution [51]. However, the negative ζ value of SAMN@TA@BSAO generates electrostatic attraction and the consequent accumulation of oppositely charged species (H_3_O^+^), causing a lower pH in the proximity of the nanomaterial surface with respect to the bulk solution. Hence, in order to shed light into the structure-function relationship and on the effect of pH on native and nanoimmobilized BSAO, both enzyme forms were characterized by circular dichroism (CD) as a function of pH. The CD spectrum of native BSAO showed a positive peak at 200 nm and a negative broad peak at 220 nm, common in proteins (Figure 6, black line) [52]. The same features were observed in the CD spectrum of the SAMN@TA@BSAO hybrid (Figure 6, blue line), providing additional evidence of the formation of the SAMN@TA@BSAO complex, as well as of the retention of the overall structure of the native enzyme.

The secondary structure composition was attained by deconvolution of the CD spectra of BSAO and SAMN@TA@BSAO by BeStSel™ (ELTE Eötvös Loránd University, Budapest, Hungary) software and are reported in Table 2. As in the case of FTIR measurements, the deconvolution process showed a high content of undetermined secondary structures in BSAO and a dominant β-sheet component, with a limited content of α-helices (≈10%). Enzyme immobilization led to a decrease in α-helix content (−70%) and an increase in β-sheet contribution (+17%), suggesting, as observed by FTIR, conformational modifications upon BSAO binding to the SAMN@TA hybrid. Other secondary structures appeared unaltered.

Aiming at unveiling pH-induced enzyme rearrangements, the secondary structure contents of the soluble and immobilized enzymes were compared using CD spectroscopy. The fraction of β-sheet over α-helix content for the two enzyme forms as a function of pH is reported in Figure 7.

In comparison to native BSAO, SAMN@TA@BSAO is characterized by a higher β-sheet to α-helix ratio in the whole examined pH range (Figure 7), suggesting a higher degree of compactness, as already postulated by FTIR and confirmed by CD measurements at neutral pH. Specifically, the CD analysis revealed a higher sensitivity to the pH variations of the secondary structure associated with the immobilized enzyme than native BSAO. Indeed, in SAMN@TA@BSAO the β-sheet to α-helix ratio showed an overall 70% increase from pH = 8.0 to pH = 4.0, while the secondary structure of native BSAO in the same pH interval was minimally influenced. Therefore, even if the observed shift in optimum pH upon enzyme immobilization is far from being fully understood, this phenomenon can possibly be linked with pH-related conformational behavior, which emerged as a distinctive tract of the immobilized enzyme. On the other hand, the pH insensitivity of native BSAO remains a matter of concern, particularly with regard to the significant decrease in the catalytic constant of the free enzyme registering below pH 6.0.

It is worth mentioning that this decrease in enzyme activity was already explained in terms of amino acid residues protonation in the catalytic site. Hereafter, in order to deepen our comprehension of the structure–function relationship of native BSAO, the possible occurrence of minor structural rearrangements was investigated by NMR spectroscopy as a function of pH.

### 2.3. ^19^F-NMR Study of BSAO as A Function of pH

The measurement of nuclear relaxation rates of specific probes has been widely studied for obtaining information on the structure of metalloenzymes [53]. In the present case, NMR spectroscopy measurements were performed at 0.42 and 7.4 T using ^1^H and ^19^F as nuclear probes. Specifically, a paramagnetic center, S, such as a transition metal, is able to induce a strong enhancement of relaxation rates of a nucleus (the probe), I, depending on I-S distance, r, according to Equations (S2)–(S9) reported in Appendix A. Therefore, relaxation rate enhancement (RRE) by paramagnetic metals, such as Cu^2+^ in BSAO, is a powerful tool for the detection of minor structural modifications of metalloproteins. In particular, longitudinal (*T*_1_^−1^) and transversal (*T*_2_^−1^) relaxation rates of ^1^H_2_O and ^19^F^−^ ion as nuclear probes were acquired at 0.42 and 7.4 T in the presence of BSAO as a function of pH, ionic strength, and temperature. The approach was already successfully used to elucidate the structure of paramagnetic sites in proteins [54,55], and the theoretical background is summarized in the Appendix A.

Table 3 shows the relaxation rates of water protons and ^19^F^−^ ion at 0.42 and 7.4 T in the presence of BSAO (*T*_1*p*_^−1^ and *T*_2*p*_^−1^) at different pH values (see Materials and Methods). Molar relaxivity of BSAO on fluoride ion, namely, the relaxation rates of ^19^F^−^ normalized for the enzyme concentration ($\frac{T_{1p}^{-1}}{\left[BSAO\right]}$ and $\frac{T_{2p}^{-1}}{\left[BSAO\right]}$), showed a pH dependence, increasing at acidic pH (Table 3).

Considering the relaxation rates of ^19^F^−^ in the presence of BSAO, at least at pH > 7.0 (Table 3), it appears that the longitudinal relaxation rate (R1, $\frac{T_{1p}^{-1}}{\left[BSAO\right]}$) depended on the main magnetic field (five times higher at 0.42 T than at 7.4 T). Transversal relaxation rate of ^19^F^−^ (R2, $\frac{T_{2p}^{-1}}{\left[BSAO\right]}$) was minimally influenced by the main field. Both relaxation rates were determined at pH 7.0 as a function of ionic strength (0.04–0.5 M KCl), and results were analyzed as per Debye–Hückel [56]. See Appendix A. Appendix A shows that the longitudinal relaxation rate of ^19^F^−^ in the presence of BSAO was insensitive to ionic strength, while the transversal relaxation rate depended on electrolyte concentration, and the slope of the log($\frac{T_{2p}^{-1}}{\left[BSAO\right]}$) vs. I^½^ curve evidenced the presence of a partial negative charge belonging to the protein (Z_BSA0_ × Z_F_- = +0.44) and interacting with F^−^ (Appendix A).

Considering Appendix A, these findings suggest that $\frac{T_{1p}^{-1}}{\left[BSAO\right]}$ was in the “fast exchange region,” i.e., *T*_1*M*_ » τ_m_. Conversely, the independence on the main field and the influence of environmental factors, such as ionic strength, indicate that $\frac{T_{2p}^{-1}}{\left[BSAO\right]}$ was in the “slow exchange region,” i.e., *T*_2*p*_ is dominated by τ_m_ (τ_m_ » *T*_2*M*_). For this reason, *T*_2*p*_ was considered less informative than the longitudinal relaxation rate for obtaining information on the interaction between Cu^2+^ in the enzyme active site and F^−^. Thus, longitudinal relaxivity of BSAO ($\frac{T_{1p}^{-1}}{\left[BSAO\right]})$ was further explored in detail as a function of pH in the 5.0–9.5 range (Figure 8). 

For comparison, the relaxation rate of ^1^H_2_O was determined as well (Figure 8), and did not appear sensitive to the functional state of the enzyme (see Figure 5). For this reason, it was not considered informative for the characterization of native BSAO. In fact, no change in longitudinal and or transversal (not shown) relaxation rate of ^1^H with respect to pH was observed in the pH range explored. This behavior was different from that of pig plasma amine oxidase, which showed a sudden doubling of the ^1^H relaxation rate above pH 8.0 [57]. Conversely, a dramatic increase in longitudinal relaxation rate of ^19^F^−^ in the presence of BSAO was observed at pH values lower than 6.0. It is worth noting that this increase of $\frac{T_{1p}^{-1}}{\left[BSAO\right]}$ of ^19^F^−^ cannot be explained by acid–base equilibria of F^−^ or BSAO. Indeed, the fitting of the experimental values with theoretical titration curves of HF or ionizable groups of the enzyme [13] was not possible (data not shown).

The effect of native BSAO on the longitudinal relaxation rate ($\frac{T_{1p}^{-1}}{\left[BSAO\right]}$) of fluoride ion was further studied as a function of temperature (20–45 °C range) and of F^−^ concentration (0.02–1.0 M) at 7.4 T. Measurements carried out at pH 7.0 as a function of temperature showed an Arrhenius dependence, and the calculated activation energy of the F^−^-Cu^2+^(BSAO) interaction was +28.5 kJ mol^−1^. The stability constant value (*K_f_*) of the ^19^F^−^-BSAO-complex was 0.49 M^−1^ at pH 7.0. The stability constant of the ^19^F^−^-BSAO complex was also determined at pH 5.2, and was *K_f_* = 2.6 M^−1^.

The results reported above allowed us to estimate the distance between the F^−^ probe and the Cu^2+^ metal center of BSAO at pH 7.0 and pH 5.2 to be 4.3 Å and 3.1 Å, respectively (see Appendix A). Thus, the increase in *K_f_* values and the decrease in Cu^2+^ − F^−^ distances under acidic pH supports the idea that the observed increase of the molar longitudinal relaxivity of F^−^ can be ascribed to the higher accessibility of the nuclear probe to the paramagnetic metal ion in the enzyme active site. This thus indicates a modification of the ligand field of Cu^2+^. Therefore, the significant enhancement in $\frac{T_{1p}^{-1}}{\left[BSAO\right]}$ of the fluoride ion can be ascribed to a localized structural modification of BSAO under acidic conditions.

Despite the fact that the activity behavior of BSAO at different pH values depends on the acid–base properties of two amino acids in the active site [15,58], the described structural alteration of the catalytic site is fully in harmony with the reported decrease in activity of the native enzyme below pH 6.0. It can be concluded that, differently from native BSAO, in which only faint conformational rearrangements (neglected by CD) can be observed at pH values far from the optimum, the structure of the nanoimmobilized enzyme revealed a higher sensitivity to pH.

## 3. Discussion

The use of enzymes as drugs is constrained by several factors, including the loss of catalytic activity and low bioavailability. Enzyme–nanomaterial hybridization can respond to these limitations and can also lead to unexpected biological properties. Herein, a functional nanodevice was developed by a self-assembled wet reaction between BSAO and SAMN@TA. Noteworthily, the enzymatic cargo was endowed with new chemical–physical and biological features as a consequence of the direct immobilization. Firstly, BSAO immobilized on SAMN@TA showed preserved catalytic activity, even if reduced in comparison to the native enzyme, accompanied by a drastic shift in pH optimum from 7.2 to 5.0. Moreover, the TA layer conferred a polyanionic character to the SAMN@TA@BSAO ternary hybrid, displaying a point of zero charge at pH = 3.5, dramatically below the isoelectric point of BSAO (pH = 6.0). Fully in harmony with the biological activity, circular dichroism spectroscopy showed a conformational alteration of the secondary structure with an increase in the β-sheet fraction upon BSAO immobilization, interpretable as an increase of protein compactness consequent to the binding, with effect on the enzymatic activity.

Furthermore, the CD analysis revealed that the immobilized enzyme possessed a pH-related plasticity, and the β-sheet to α-helix ratio showed an overall 70% increase going from pH 8.0 to pH 4.0. Conversely, the secondary structure of native BSAO was minimally influenced in the same pH interval. The pH stability of the soluble enzyme was further confirmed by an in-depth NMR investigation indicating that the reported decrease of activity below pH 6 can be influenced by localized structural modifications of the catalytic site, not perceivable by circular dichroism or infrared spectroscopy.

Regarding the behavior of catalytic activity on spermine, the soluble and immobilized enzymes responded to pH in opposite ways. Native BSAO was highly sensitive to pH due to the protonation of residues in the active site controlling the interaction with substrate, whereas the SAMN@TA@BSAO form was not significantly responsive to pH variations, showing its optimum catalytic activity at pH 5.0, two pH units below the native enzyme. This behavior suggests that the microenvironment of BSAO immobilized on TA-coated nanoparticles represents the most important factor controlling its activity.

In conclusion, along with the properties of the nanomaterial core, combining superparamagnetism and intrinsic fluorescence [59], the presented ternary hybrid can be ideally exploited for cancer therapy, as many cancerous tissues are characterized by polyamine overproduction and acidic pH of the tumor environment. Overall, the present work enriches the current awareness about the often-unpredictable advantages derived by enzyme–nanoparticle hybridization.

## 4. Materials and Methods

### 4.1. Chemicals

All reagents were purchased at the highest commercially available purity and were used without further purification processes. Bovine serum amine oxidase (amine: O_2_ oxidoreductase, deaminating, E.C. 1.4.3.6) was purified according to [7]. Tannic acid, spermine, 4-amino-antipyrine, horseradish peroxidase type II (HRP, 179 U/mg), N,N-dimethylaniline were obtained from Sigma-Aldrich/Merck (St. Louis, MO, USA). Water for sample preparation and analysis was obtained from a Genie Direct-Pure (RephiLe Bioscience Ltd., Shanghai, China) ultrapure water device with a resistivity of at least 18 MΩ cm^−1^. A series of Nd-Fe-B magnets (N35, 263–287 kJ/m3 BH, 1170–1210 mT flux density by Power magnet—Germany) was used to magnetically recover the particles.

### 4.2. Instrumentation

In order to determine enzyme purity, copper content of BSAO was calculated by integration of the electron spin resonance (ESR) spectrum against Cu^2+^–EDTA standard. A ratio of 2.0 mol of Cu^2+^/mol enzyme was found. The EPR spectra were acquired by a Bruker D-200 spectrometer operating at 9 GHz. Transmission electron microscopy (TEM) images were acquired by a JEOL JEM-2010 microscope (JEOL Ltd., Tokyo, Japan) operating at 200 kV with a point-to-point resolution of 1.9 Å. Copper-mesh TEM grids were purchased from SPI Supplies, West Chester, PA, USA. Before TEM measurements, samples were dispersed in ethanol and the suspension was treated by ultrasound for 10 min. A drop of dilute suspension was placed on a carbon-coated copper grid and allowed to dry by evaporation at room temperature. X-ray diffraction (XRD) analyses were carried out using a Rigaku Oxford Diffraction SuperNova single-crystal diffractometer equipped with a 200 K Dectris detector operating with a microsource Mo Kα X-ray radiation at a wavelength of 0.71073 Å. All samples were glued on top of glass fibers and measured in the range of 0–75° with a step size of 1° around the φ-axis and 20 s per-frame acquisition time. The data were processed using CrysAlisPro 40_64.67a and Panalytical HighScore Plus software. Scanning electron microscopy–energy-dispersive spectroscopy (SEM-EDS) analyses were performed employing a Tescan Solaris scanning electron microscope operating at 10 and 1 KeV. Samples were air dried on a copper grid. EDS measurements were collected for 30 s.

Dynamic light-scattering measurements and zeta potential of SAMNs, SAMN@TA, and SAMN@TA@BSAO hybrids were measured by a Zetasizer Nano particle analyzer ZEN3600 (Malvern Instruments, Malvern, UK). Circular dichroism (CD) analysis of BSAO, SAMN@TA, and SAMN@TA@BSAO were carried out using an N_2_ flushed JASCO J-810 spectropolarimeter using a 2 mm quartz cuvette (Hellma Analytics). Measurements were performed in aqueous solutions containing 2 mM Britton–Robinson buffer and 25 mM KCl. Optical spectroscopy measurements were performed in a 1 cm light path quartz cuvette (Hellma Analytics) using a Cary 60 (Agilent Technologies) UV-visible spectrophotometer. Fourier-transform infrared (FT-IR) analysis of native enzymes, bare SAMNs, SAMN@TA, and SAMN@TA@BSAO were recorded using an IR Affinity-1S spectrometer (Shimadzu Corp., Kyoto, Japan) equipped with a diamond ATR analyzer and LabSolutions IR software (Shimadzu Corp., Kyoto, Japan). The scanning range was between 500 and 4000 cm^−1^ with a resolution of 4 cm^−1^ and 300 scans. The quantitative analysis of secondary structure of native enzyme and SAMN@TA@BSAO hybrid was based on a curve fitting of amide I band according to Hebia and coauthors [42]. Secondary structure content was quantified by band deconvolution using a Gaussian model considering the following secondary structure motifs: β-sheet (1637–1610 cm^−1^), random coil (1648–1638 cm^−1^), α-helix (1660–1650 cm^−1^), β-turn (1680–1660 cm^−1^) and β-antiparallel (1692–1680 cm^−1^). Nuclear magnetic relaxation measurements of ^1^H and ^19^F were carried out at 0.42 T (16 MHz proton Larmor frequency) with a homemade pulse spectrometer. Fluorine, ^19^F, relaxation measurements were also carried out at 7.4 T (300 MHz proton Larmor frequency) with a 300 MSL Bruker apparatus. Spin lattice relaxation times (*T*_1_) were measured by saturation recovery pulse sequence at 0.42 T or by inversion recovery sequence at 7.4 T. The transverse relaxation times (*T*_2_) were measured by the Meiboom–Gill modification of the Carr–Purcell spin-echo experiment. Values of paramagnetic ion contribution to the relaxation rates of nuclear probes (*T*_1*p*_^−1^ and *T*_2*p*_^−1^) were obtained by subtracting from *T*_1_^−1^ and *T*_2_^−1^ values those of the corresponding relaxation rates measured in diamagnetic buffer solutions. T_1*p*_^−1^ and *T*_2*p*_^−1^ were normalized for the concentration of paramagnetic species by dividing by [*BSAO*] ($\frac{T_{1p}^{-1}}{\left[BSAO\right]}$ and $\frac{T_{2p}^{-1}}{\left[BSAO\right]}$). These normalized values represent the molar relaxivity, R, of BSAO, which is the relaxation rate induced per mole of protein per liter. The stability constant (*K_f_*) of the BSAO-^19^F^−^ complex under the [*F*^−^] >> [*BSAO*]_tot_, conditions ([*BSAO*]_tot_ is the total enzyme concentration) was calculated according to the following equation [60]:(3)$$T_{1p}=\frac{A}{\left[BSAO\right]}\left(\frac{1}{K_{f}}+\left[F^{-}\right]\right)$$
where *A* is a graphical constant (expressed in s), *K_f_* is the stability constant of the BSAO-^19^F^−^ complex, and [*BSAO*] and [*F**^−^*] are the enzyme and fluoride ion concentrations, respectively. According to this equation, the measurements of relaxation rates of ^19^F^−^ in the 0.02–1.0 M concentration range in the presence of BSAO showed that the *T*_1*p*_ vs. [*F*^−^] plot was linear, and K_F_ was calculated from the intercept on the *x*-axis. Transition-metal contaminants eventually present in protein solutions were removed by dialyzing enzyme aliquots against buffer containing 100 µM EDTA. Glassware was thoroughly washed with concentrated HCl solution (2 M) before use.

### 4.3. Preparation of SAMN@TA and SAMN@TA@BSAO Hybrids

SAMNs (surface-active maghemite nanoparticles) were produced in-house and their synthesis and extensive characterization is reported elsewhere [61]. The core–shell hybrid nanomaterial (SAMN@TA) constituted of SAMNs (core) and tannic acid (TA, shell) was prepared as follows: a 0.5 g L^−1^ colloidal suspension of naked SAMNs was incubated with freshly prepared 100 μM tannic acid in 50 mM tetramethylammonium perchlorate (N(CH_3_)_4_ClO_4_) solution at pH 7.1 at room temperature under overnight rotating mixing using a FALC F205 rotary shaker (FALC Instruments, Treviglio, Italy). After the incubation period, the nanoparticles were separated by application of external magnetic field and extensively washed with 20 mM HEPES buffer at pH 7.2 to remove unbound tannic acid. The amount of TA bound on SAMNs was calculated from the disappearance of the TA absorbance at 280 nm in the supernatants using the molar extinction coefficient of 38.5 × 10^3^ M^−1^cm^−1^, as already described elsewhere [25] The SAMN@TA@BSAO ternary hybrid was prepared as follows: a colloidal suspension of SAMN@TA (0.5 g L^−1^) was incubated with 200 mg L^−1^ BSAO in HEPES buffer 20 mM at pH 7.2 for 2 h at 4 °C under rotating mixing. After incubation, the hybrid was recovered using an external magnetic field and washed several times with HEPES buffer 20 mM at pH 7.2. The concentration of immobilized BSAO onto SAMN@TA was determined by UV-vis spectroscopy as per Stevanato [62].

### 4.4. Kinetic Characterization of BSAO and SAMN@TA@BSAO Hybrid

The enzyme activity of soluble BSAO and of the SAMN@TA@BSAO hybrid were carried out in 10 mM Britton–Robinson buffer [62] at constant ionic strength and pH values of 4.0 to 8.0, using increasing concentrations of spermine as substrate. The pH was adjusted by addition of HCl and NaOH. The enzyme activity was determined by measuring the H_2_O_2_ generation rate according to a spectrophotometric method previously reported, requiring horseradish peroxidase (HRP) and a reduced dye. Specifically:(4)$$R-CH_{2}NH_{2}+O_{2}+H_{2}O\;\overset{Amine\;oxidase}{\to }\;R-\mathrm{CHO}+NH_{3}+H_{2}O_{2}$$
(5)$$H_{2}O_{2}+Dye_{red}\;\overset{HRP}{\to }\;H_{2}O_{2}+Dye_{ox}$$

According to the assay described by Stevanato et al. [63] the reaction rate of second reaction (Equation (5)) is always higher than that of the first reaction (Equation (4)).

These measurements were performed following the increase in absorbance of the HRP-catalyzed secondary reaction product at 555 nm using a molar extinction coefficient of 1.25 × 10^4^ M^−1^ cm^−1^. All measurements were carried out at room temperature (22 ± 1 °C). Kinetic parameters (*k_cat_* and *K_M_*) were calculated by fitting the Michaelis–Menten equation to the experimental data:
(6)$$v=\frac{k_{cat}\left[BSAO\right]\left[spermine\right]}{K_{M\;}+\left[spermine\right]}$$
where v is the reaction rate in M s^−1^, *k_cat_* is the catalytic constant, [*BSAO*] is the enzyme concentration, [*spermine*] is substrate concentration, and *K_M_* is the Michaelis constant [49]. In Appendix A, examples of the reaction rates of native BSAO and of the SAMN@TA@BSAO complex are shown. OriginLab software (version 7.5; OriginLab Corporation, Northampton, MA, USA) was used for the analysis of kinetic data. The kinetic constants were calculated by assuming an enzyme molecular weight of 180 kDa [7,8].

## Figures and Tables

**Figure 1 ijms-23-12172-f001:**
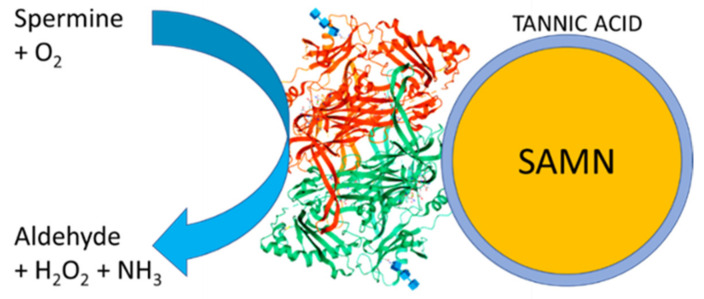
Schematic representation of BSAO enzyme–nanoparticle hybrid (i.e., SAMN@TA@BSAO) catalyzed oxidation of a polyamine (i.e., spermine) to cytotoxic products—aldehyde, ammonia, and hydrogen peroxide—in the presence of oxygen.

**Figure 2 ijms-23-12172-f002:**
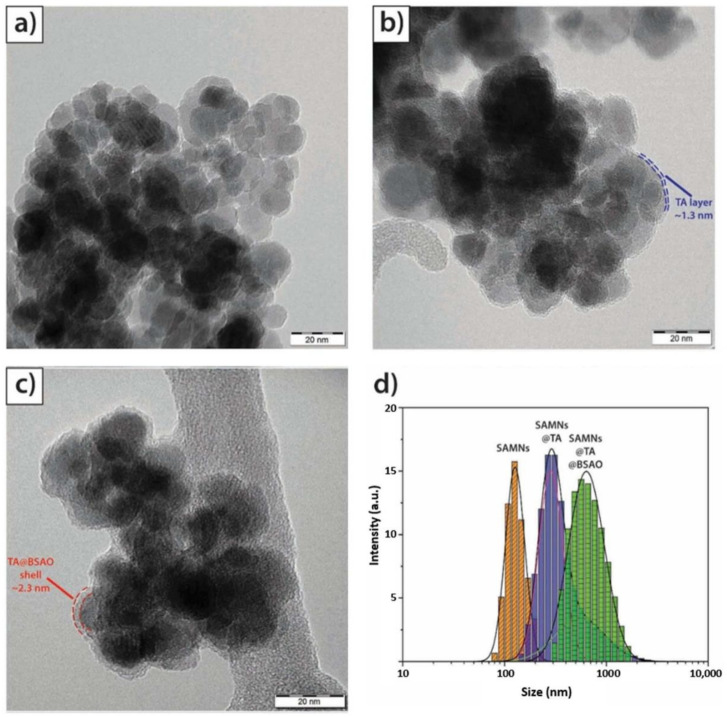
TEM and DLS analyses of the naked SAMNs, SAMN@TA, and SAMN@TA@BSAO hybrids. (**a**) TEM image of SAMNs; (**b**) TEM image of SAMN@TA; (**c**) TEM image of SAMN@TA@BSAO; (**d**) hydrodynamic sizes of naked SAMNs, SAMN@TA, and SAMN@TA@BSAO.

**Figure 3 ijms-23-12172-f003:**
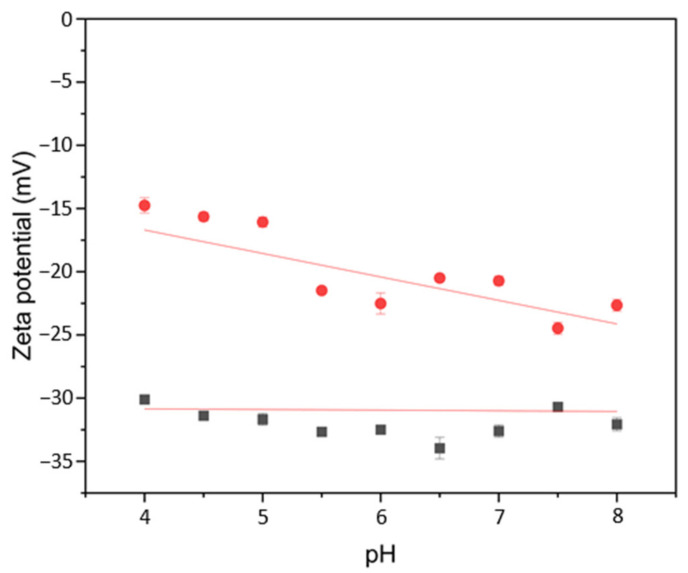
Zeta-potential values of the SAMN@TA and of the SAMN@TA@BSAO ternary hybrid as a function of pH. (■) SAMN@TA; (●) SAMN@TA@BSAO.

**Figure 4 ijms-23-12172-f004:**
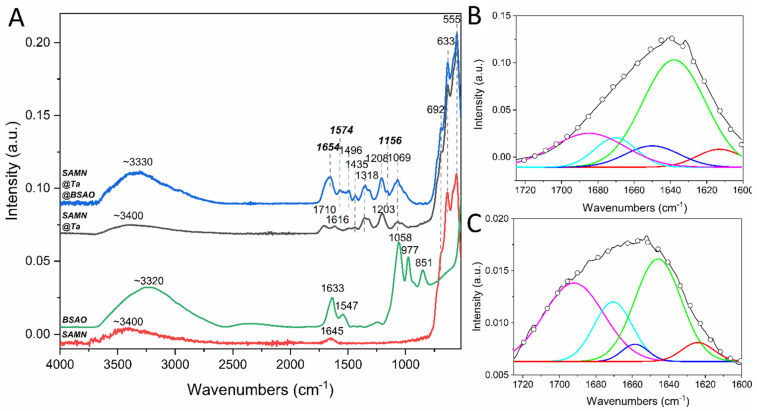
FTIR spectra of bare SAMNs, BSAO, SAMN@TA and SAMN@TA@BSAO. (**A**)—infrared spectra of SAMN (red line), BSAO (green line), SAMN@TA (black line) and SAMN@TA@BSAO (blue line). Deconvolutions of: (**B**)—BSAO and SAMN@TA@BSAO; (**C**)—amide I vibrational band; experimental amide I (black line), amide I gaussian curve fitting (dotted), β-sheet (red line), random coil (green line), α-helix (blue line), β-turn (light blue line) and β-antiparallel (purple line). Measurements were carried out in water.

**Figure 5 ijms-23-12172-f005:**
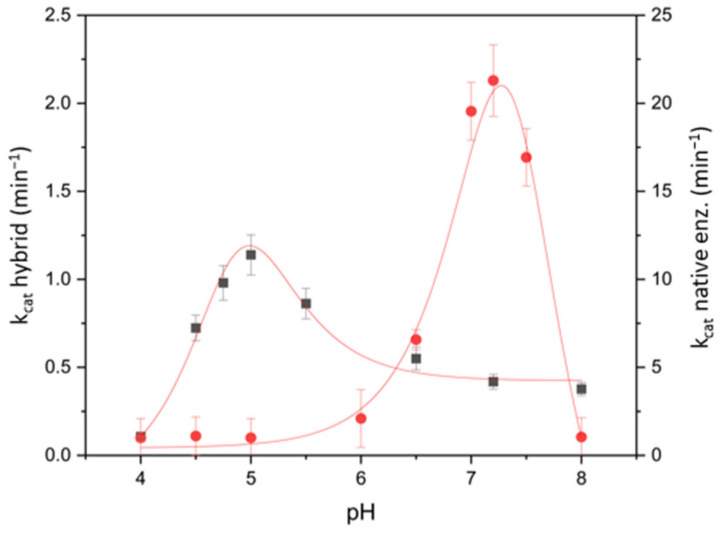
Catalytic constant, *k_cat_*, of the oxidation of spermine by native BSAO and SAMN@TA@BSAO. The left *Y*-axis reports the *k_cat_* values for the SAMN@TA@BSAO hybrid, while the right *Y*-axis reports *k_cat_* values that refer to native BSAO. Curves were fitted according to equations reported in Appendix A. (■) SAMN@TA@BSAO; (●) BSAO.

**Figure 6 ijms-23-12172-f006:**
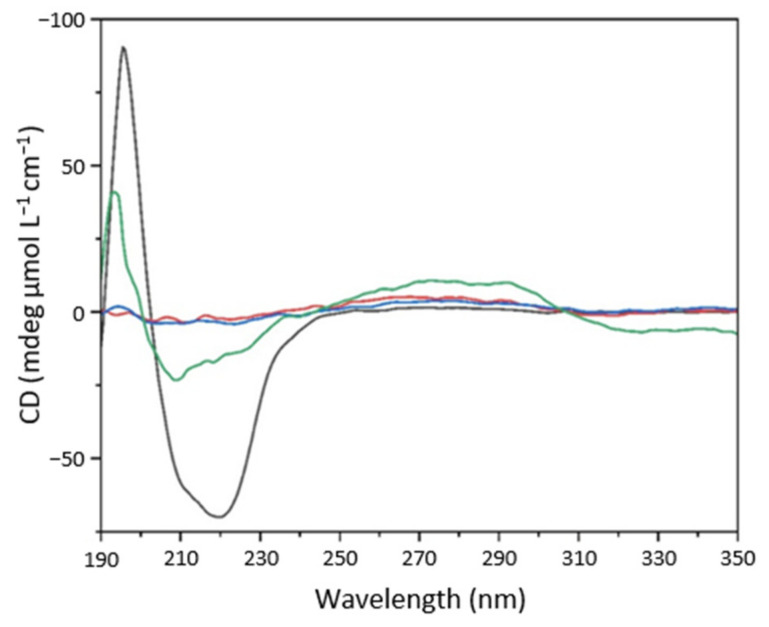
Circular dichroism spectra of BSAO (black), naked SAMNs (red), SAMN@TA (blue), and SAMN@TA@BSAO (green).

**Figure 7 ijms-23-12172-f007:**
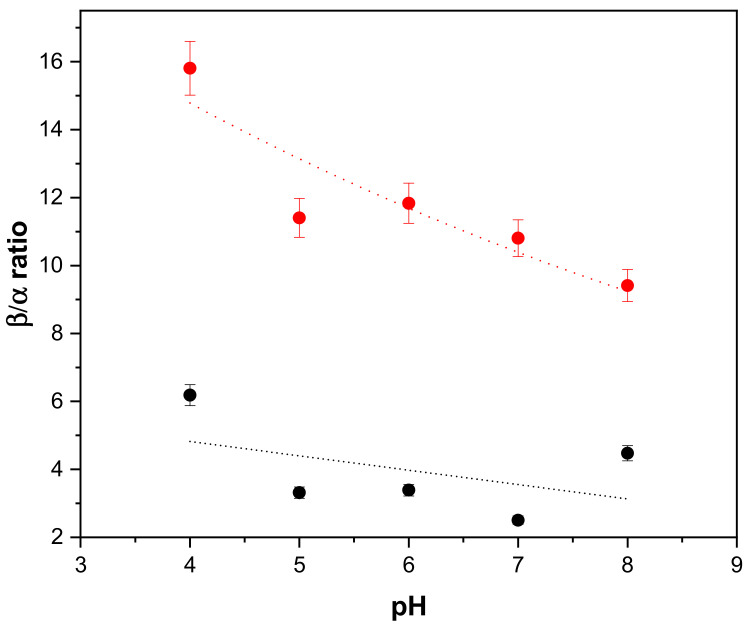
β-sheet to α-helix ratio of native BSAO and SAMN@TA@BSAO as a function of pH. Measurements were carried out by CD in 2 mM Britton–Robinson buffer with 25 mM KCl. (●) 0.83 µM BSAO; (●) 0.5 mg/mL SAMN@TA@BSAO.

**Figure 8 ijms-23-12172-f008:**
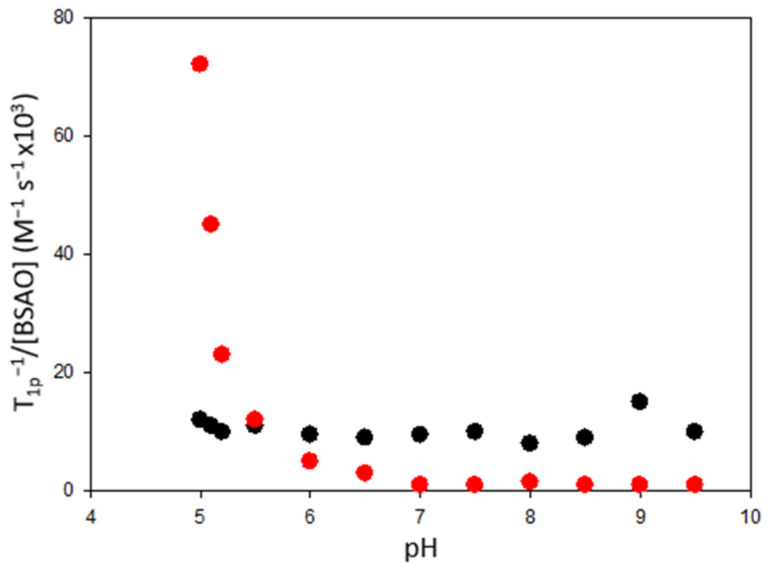
Effect of pH on longitudinal relaxation rates ^19^F^−^ and ^1^H_2_O in the presence of BSAO. A buffer containing 33 mM citric acid, 44 mM imidazole, and 44 mM AMPSO (i.e., N-(1,1-dimethyl-2-hydroxyethyl)-3-amino-2-hydroxypropanesulfonic acid) at constant ionic strength (0.51 M) was used. Longitudinal relaxation rates: water protons, ^1^H (●) and ^19^F^−^ (●). Measurements were performed at 0.42 T. KF (0.5 M) was added to the buffer, while the concentration of BSAO was in the range 30–600 µM. Measurements were performed at 22 ± 1 °C.

**Table 1 ijms-23-12172-t001:** Secondary structure contents of native BSAO and of the SAMN@TA@BSAO hybrid according to the deconvolution of amide I band of the FTIR spectra.

	α-Helix (%)	β-Sheet (%)	β-Turn (%)	Others (%)
Native BSAO	10 ± 5	27 ± 3	10 ± 1	53 ± 3
SAMN@TA@BSAO	5 ± 1	32 ± 2	11 ± 1	52 ± 4

**Table 2 ijms-23-12172-t002:** Secondary structure contents of native BSAO and of the SAMN@TA@BSAO hybrid according to circular dichroism spectra at pH 7.2.

	α-Helix (%)	β-Sheet (%)	β-Turn (%)	Others (%)
Native BSAO	10 ± 1	29 ± 2	12 ± 1	49 ± 3
SAMN@TA@BSAO	3 ± 1	34 ± 2	13 ± 2	50 ± 4

**Table 3 ijms-23-12172-t003:** Longitudinal and transversal relaxation rates of fluoride ion and water protons in the presence of native BSAO measured at 0.42 and 7.4 T. Molar relaxivity values ($\frac{T_{1p}^{-1}}{\left[BSAO\right]}$, R1 and $\frac{T_{2p}^{-1}}{\left[BSAO\right]}$, R2) were measured at 22 ± 1 °C, at constant ionic strength (0.51 M).

	Magnetic Field (Tesla)					
pH	0.42 T				7.4 T	
	*^19^F* ^−^		*^1^H_2_O*		*^19^F* ^−^	
	R1 (M^−1^s^−1^) × 10^−4^	R2 (M^−1^s^−1^) × 10^−4^	R1 (M^−1^s^−1^) × 10^−4^	R2 (M^−1^s^−1^) × 10^−4^	R1 (M^−1^s^−1^) × 10^−4^	R2 (M^−1^s^−1^) × 10^−4^
5.2	2.3	7.2	1.0	2.7	2.3	7.6
7.0	0.53	5.2	0.95	1.4	0.1	4.8
9.0	0.50	2.6	1.5	2.9	0.1	3.4

## Data Availability

The original data are available upon reasonable request to the corresponding author.

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
