# Peer review of "Acidic Shift of Optimum pH of Bovine Serum Amine Oxidase upon Immobilization onto Nanostructured Ferric Tannates"

_ijms, 2022, doi:10.3390/ijms232012172_

Round 1

Reviewer 1 Report

I read the paper carefully and with interesting. 

The introduction section must be improved. Authors must explain about Tannic Acid in more details. Authors should cite and use this ref: https://doi.org/10.1016/j.cej.2021.134146

Another thing is about nanomaterials characterizations. Characterizations are insufficient. Authors must provide other analysis like FTIR, SEM-EDX, XRD and TGA

Author Response

Point 1. The introduction section must be improved. Authors must explain about Tannic Acid in more details. Authors should cite and use this ref: https://doi.org/10.1016/j.cej.2021.134146.

Answer 1. A paragraph highlighting the general importance of TA-based materials was introduced in the revised version of the manuscript (please, see page 2 of the revised text). In particular, the suggested review paper was cited. Authors are grateful to the reviewer for this contribution.

Point 2. Another thing is about nanomaterials characterizations. Characterizations are insufficient. Authors must provide other analysis like FTIR, SEM-EDX, XRD and TGA.

Answer 2. An extensive FTIR characterization is already present in the submitted article. Additional chemical-physical analyses of the nanomaterials were introduced in the revised version of the manuscript. In particular, SAMN@TA and SAMN@TA@BSAO were compared to bare SAMNs by XRD and SEM-EDX, demonstrating the conservation of the maghemite core and further confirming the formation of the carbonaceous shells (see page 4 of the revised text and new Figures S2 and S3 in Supplementary Materials). The numbering of SI figures was checked and corrected.

Reviewer 2 Report

1) There was a mix up in the use of the enzyme BSAO. For instance, lines 19, and 33 said Bovine Serum AMINE Oxidase, while line 45 said, AMINE Oxidase but line 51 referred it as Bovine Serum AMINO Oxidase while line 460 used Bovine PLASMA Amine Oxidase.

There is need for clarity as there is a difference between serum and plasma and also between amine group and amino group 

2) lines 57-58 tends to repeat lines 59-60, only that the later is more specific

3) lines 82-88 fits in as results rather than introduction. authors should rather take a critical look at the statement and position it appropriately with its accompanying fig 1

4)  lines 471 should have a full name of ESR- in bracket after the abbreviation ESR-spectrum 

5)  line 511; the equation 3 lacked keys to identify the symbols used. moreso, the value of the constant "A" was not stated

6) lines 521-525 and that of lines 531-532 were done overnight under rotating mixing. the name and model of the equipment used to carry out the mixing should be stated clearly

7) line 527; the pH of the 20mM HEPES buffer should be stated

8) line 528; the amount of TA bound on SAMNs was calculated from the disappearance of TA absorbance at 280nm in the supernatants is a very cloudy statement. Further information should be provided for reproducibility.  

9) lines 541-543; HRP is an enzyme which is being used to assay other enzymes like the BSAO and its hybrid. the methodology should be clearly stated to enable reader be clear on which activity that is being measured and there is conflict with activity of HRP

10) LINES 546-549;  the graph supporting the statement should be shown clearly in the result section and appropriate reference made to it

Author Response

Reviewer 2:

Point 1.  There was a mix up in the use of the enzyme BSAO. For instance, lines 19, and 33 said Bovine Serum AMINE Oxidase, while line 45 said, AMINE Oxidase but line 51 referred it as Bovine Serum AMINO Oxidase while line 460 used Bovine PLASMA Amine Oxidase. There is need for clarity as there is a difference between serum and plasma and also between amine group and amino group.

Answer 1. The authors agree with the Reviewer. The correct enzyme is Bovine Serum Amine Oxidase and the text was modified accordingly.

Point 2. lines 57-58 tends to repeat lines 59-60, only that the latter is more specific.

Answer 2. The text was modified as suggested (please, see page 2 of the revised text).

Point 3. lines 82-88 fits in as results rather than introduction. authors should rather take a critical look at the statement and position it appropriately with its accompanying fig 1.

Answer 3. We agree with reviewer: the statement was misleading. Actually, the paragraph was aimed at describing the biological activity of the nano-bio-conjugate and, therefore, introducing the experimental work. The sentence was rephrased and moved, as well as Figure 1,  to the Results section  as suggested (please see page 3 of the revised manuscript).

Point 4. lines 471 should have a full name of ESR- in bracket after the abbreviation ESR-spectrum

Answer 4. Electron spin resonance was enunciated in the revised text as suggested. We thank the reviewer.

Point 5. line 511; the equation 3 lacked keys to identify the symbols used. More so, the value of the constant "A" was not stated.

Answer 5. Symbols and the value of the "A" constant were defined as requested (please, see page 15 of the revised text).

Point 6. lines 521-525 and that of lines 531-532 were done overnight under rotating mixing. the name and model of the equipment used to carry out the mixing should be stated clearly

Answer 6. The information on the equipment was added to the revised text as suggested.

Point 7. line 527; the pH of the 20mM HEPES buffer should be stated.

Answer 7. The pH value was specified as requested.

Point 8. line 528; the amount of TA bound on SAMNs was calculated from the disappearance of TA absorbance at 280nm in the supernatants is a very cloudy statement. Further information should be provided for reproducibility. 

Answer 8. The preparation of the SAMN@TA complex was already extensively described. The quantification was carried out by using the reported extinction coefficient. These information were mentioned in the revised manuscript (please see page 15 of the revised manuscript).

Point 9. lines 541-543; HRP is an enzyme which is being used to assay other enzymes like the BSAO and its hybrid. the methodology should be clearly stated to enable reader be clear on which activity that is being measured and there is conflict with activity of HRP

Answer 9. The methodology used for the measurement of BSAO activity was clearly described and the specific reference was highlighted (please see page 15 of the revised manuscript).

Point 10. LINES 546-549; the graph supporting the statement should be shown clearly in the result section and appropriate reference made to it

Answer 10. The classical model for describing enzyme kinetics was added to the revised text as suggested. A new figure presenting the reaction rate of native BSAO and of the SAMN@TA@BSAO complex was introduced in the Supplementary Materials (please see Figure S6).
